# Immunotherapeutic potential of collagen V oral administration in mBSA/CFA-induced arthritis

**Lizandre Keren Ramos da Silveira**[1], **Ana Paula P. Velosa**[1], **Sergio Catanozi**[2], **Marco Aurélio A. Pereira**[3], **Antonio dos Santos Filho**[1], **Fabio Luiz N. Marques**[4], **Daniele de Paula Faria**[4], **Caroline Cristiano Real**[4], **Sandra de M. Fernezlian**[5], **Amanda Flores Yanke**[1], **Zelita Aparecida de J. Queiroz**[1], **Vitória Elias Contini**[1], **Thays de Matos Lobo**[1], **Solange Carrasco**[1], **Camila Machado Baldavira**[5], **Cláudia Goldenstein-Schainberg**[1], **Ricardo Fuller**[1], **Vera L. Capelozzi**[5], **Walcy R. Teodoro**[1] *

1 Division of Rheumatology, Faculdade de Medicina da Universidade de Sao Paulo, Sao Paulo, SP, Brazil, 2 Laboratorio de Lipides (LIM-10), Hospital das Clinicas (HCFMUSP) da Faculdade de Medicina da Universidade de Sao Paulo, Sao Paulo, Brazil, 3 Department of Surgery, School of Veterinary Medicine and Animal Science, University of Sao Paulo, Sao Paulo, Brazil, 4 Laboratory of Nuclear Medicine (LIM 43), Hospital das Clinicas HCFMUSP, Faculdade de Medicina, Universidade de Sao Paulo, Sao Paulo, SP, Brazil, 5 Department of Pathology, Faculdade de Medicina FMUSP, Universidade de Sao Paulo, Sao Paulo, São Paulo, Brazil

* walcy.teodoro@fm.usp.br

**Data Availability Statement:** The datasets used and/or analyzed during the current study are

## Abstract

We hypothesized that after synovial injury, collagen V (Col V) expose occult antigens, and Col V autoantibodies develop, indicating the loss of immune tolerance against this molecule, thus leading to damage to mesenchymal-derived cells as well as the extracellular matrix in experimental arthritis. Thus, the present study investigated the effects of oral administration of Col V on the synovium after the development of inflammation in mBSA/CFA-induced arthritis. After fourteen days of intraarticular administration of mBSA, 10 male Lewis rats were orally administered Col V (500 μg/300 μL) diluted in 0.01 N acetic acid (IA-Col V group). The arthritic group (IA group, n = 10) received only intraarticular mBSA. An intra-articular saline injection (20 μL) was given to the control group (CT-Col V, n = 5). IA group presented damaged synovia, the expansion of the extracellular matrix by cellular infiltrate, which was characterized by T and B lymphocytes, and fibroblastic infiltration. In contrast, after Col V oral immunotherapy IA-Col V group showed a significant reduction in synovial inflammation and intense expression of IL-10+ and FoxP3+ cells, in addition to a reduction in Col V and an increase in Col I in the synovia compared to those in the IA group. Furthermore, an increase in IL-10 production was detected after IA-Col V group spleen cell stimulation with Col V *in vitro*. PET imaging did not differ between the groups. The evaluation of oral treatment with Col V, after mBSA/CFA-induced arthritis in rats, protects against inflammation and reduces synovial tissue damage, through modulation of the synovial matrix, showing an immunotherapeutic potential in inhibiting synovitis.

available in the manuscript and supporting information.

**Funding:** "This work was supported by grants from the São Paulo Research Foundation - FAPESP [2019/24178-0] for the Silveira LKR, FAPESP [2018/20403-6 and 2023/02755-0] for Capelozzi, VL and Baldavira CM, FAPESP [2021/13220-5] for Velosa AP and from CNPq - National Council for Scientific and Technological Development [303735/2021-0] for Capelozzi, VL. The FAPESP is a public institution that promotes academic research of the government of the state of São Paulo, Brazil. The CNPq is an organization linked to the Ministry of Science, Technology and Innovation to encourage research in Brazil. The funders had no role in study design, data collection and analysis, decision to publish, or preparation of the manuscript".

## Introduction

Collagen V (Col V) is a glycoprotein accounting for 2 to 5% of all fibrillar collagen types. Collagen I (Col I), III (Col III) and Col V aggregate to form fibrillar structures, which are stabilized by intermolecular cross-links, resulting in the formation of a stable network, forming heterotypic fibrils (I/III/V) [1–6]. The triple helix of Col V, formed by two alpha 1 chains ($\alpha$1) and one alpha 2 chain ($\alpha$2), is hidden between the Col I and Col III fibers. Unlike other types of collagen, Col V retains its amino terminal domain, which projects to the surface of the heterotypic fiber [5, 7]. As these domains reach a critical concentration, the aggregation of new collagen molecules to the surface becomes less favorable than the formation of new fibrils [5, 7]. The Col V is essential for the formation of the collagen network, as well as the control of fibrogenesis and the regulation of fiber size [6, 8–12]. Furthermore, to its function of controlling the diameter of heterotypic fibers, Col V has other biological properties and in this sense, it has been demonstrated that a culture medium rich in Col V inhibits the growth and proliferation of endothelial cells [13, 14]. In addition, it contributes to the interaction between stromal collagen and the basal membrane and is highly important for cell adhesion and matrix repair [15]. Damage to mesenchymal-derived cells and the extracellular matrix, as in the case of synovia, can typically expose occult antigens in Col V, making it immunologically reactive. This results in synovial repair, as previously reported in pulmonary clinical studies [15, 16] and confirmed by preclinical studies [8, 17–24].

Oral immunotherapy is widely used to attenuate the course of allergies and autoimmune diseases. Studies performed in experimental models have shown that oral or nasal administration of Col V can weaken the humoral and cellular immune response, preserving lung tissue in fibrosis related to lung transplant rejection [17] and in bleomycin-induced pulmonary fibrosis [16]. Therefore, Col V has received widespread attention in recent years as a novel candidate for immunotherapy strategies [23].

In a previous study, we considered Col V a potential antigen for arthritis pathogenesis. In this way, prophylactic oral administration of Col V in an mBSA/CFA-induced arthritis model inhibited lymphocyte CD3 and CD20 subpopulations, macrophages and proinflammatory cytokines in synovial tissue [25]. However, to date, the main challenge is to clarify the effect of oral administration of Col V during experimental arthritis. For the studies reported here, we adopted the hypothesis that Col V under injury conditions exposes occult antigens with the development of Col V autoantibodies, indicating the loss of immune tolerance against this molecule and contributing to damage to mesenchymal-derived cells as well as the synovial extracellular matrix in experimental arthritis. Therefore, in the present study, conducted in rats injured by intraarticular injection of mBSA/CFA, we evaluated the potential of immunotherapy with Col V to reduce inflammatory process and synovial remodeling in a arthritis model.

We found that oral administration of Col V post intraarticular injury attenuated the inflammatory response in the synovia and accelerated the repair/remodeling of injured synovia by Col I, which was mediated by an increase in *in situ* FoxP3+ and IL-10+ in the synovial compartment; thus we suggest Col V is a potential target for arthritis treatment.

## Materials and methods

### Experimental protocol

Twenty-five three-month-old male Lewis rats with an average body mass of 360g were provided by the animal facility of our institution. All of the experimental procedures were approved by The Committee on Ethical Use of Laboratory Animals of the Faculty of Medicine

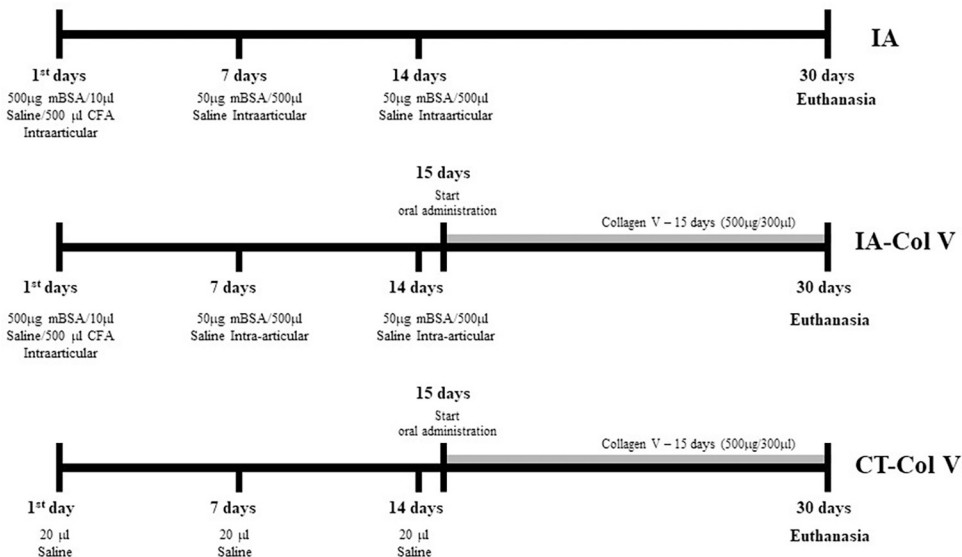

**Fig 1. Experimental protocol.** Schematic drawing of mBSA/CFA-induced arthritis and Col V oral administration.

at the University of São Paulo (Process code: 1294/2019). The experimental study followed relevant guidelines, such as ARRIVE, and global regulations.

For preventive analgesia, tramadol hydrochloride injections (40 mg/kg body weight) were administered subcutaneously one hour before anesthesia and every 8 hours for 48 hours after the experimental procedure. Intraperitoneal anesthesia (ketamine hydrochloride at a dose of 100 mg/kg body mass and xylazine hydrochloride at a dose of 10 mg/kg body mass) was administered. Then, arthritis (IA, induced arthritis; n = 20) was induced on the right knee with 20 μL of a 500 μg emulsion of mBSA (Sigma Chemical, St. Louis, MO), diluted in 10 μL of saline and 500 μL of Freund's complete adjuvant, delivered intra-articularly, followed by an intra-articular booster of 20 μL of an emulsion of 50 μg of mBSA in saline on days 7 and 14 of the experimental protocol. After the intra-articular injections, some animals (n = 10) received Col V (500 μg/300 μL) [25] diluted in acetic acid (0.01 N) by gavage 3 times a week from days 15 to 30 (IA-Col V). The control group (CT-Col V, n = 5) was given saline (20 μL) and Col V by gavage using the same protocol (Fig 1). The animals were euthanized by intraperitoneal injection of an anesthetic overdose (ketamine hydrochloride at a dose of 300 mg/kg body mass and xylazine hydrochloride at a dose of 30 mg/kg body mass).

## Histological and synovial score analysis

The synovial tissue samples were stained with hematoxylin and eosin (H&E) for morphometric analysis of the synovial architecture and inflammatory cells. Synovial tissue was analyzed at 200x magnification by two observers who did not have prior knowledge of the treatments. Krenn´s score was used, which consists of scoring the synovial lining cell layer, the density of the resident cells and inflammatory infiltrate (0–3 points), after which the points are added together to determine the degree of synovitis (0–9 points) [26].

## Evaluation of the inflammatory response using [18F]FDG PET imaging

The animals underwent a PET scan at baseline (one day before the start of the protocol), 14 days (one day before the start of Col V administration) and 30 days (at the end of the protocol). The animals were then anesthetized with 1.5%-3% isoflurane in 100% oxygen and received an intravenous

injection of the [$^{18}$F]FDG radiotracer (18–37 MBq) via the penile vein. After injection, the animals were allowed to wake up for better radiotracer distribution. After 45 minutes, the animals were again anesthetized and positioned with their knees in the center of the field of view on a PET scanner (Triumph® II Trimodality System, CA, USA). The images were reconstructed using the 3D-OSEM algorithm using 20 iterations and 4 subsets and quantified using PMOD software version 4.0. [$^{18}$F]FDG uptake is expressed as the standardized uptake value (SUV), calculated as SUV = radioactivity concentration (Bq/mL)/[injected activity (Bq)/animal weight (g)].

## Immunohistochemical analysis of inflammatory cells

Sections were deparaffinized and blocked in a 0.3% hydrogen peroxide solution to inhibit endogenous peroxidase activity (CD4, CD8 and CD20). For immunostaining of CD3 and FoxP3, the sections were blocked with 0.3% hydrogen peroxide with methanol (v/v), with a cycle of 2 washes for 10 minutes each. For immunostaining of IL-10, the sections were blocked with 0.3% hydrogen peroxide and subjected to additional protein blocking. For immunostaining of CD68, the sections were blocked with 0.3% hydrogen peroxide in methanol (v/v) and subjected to additional protein blocking.

The primary antibodies used were mouse monoclonal anti-CD3 (1:1000; 1 mg/mL; Dako), anti-CD4 (1:50; 0.2 mg/mL; Santa Cruz Biotechnology Inc.), anti-CD8 (1:50; 0.2 mg/mL; Santa Cruz Biotechnology Inc.), anti-CD20 (1:600; 0.2 mg/mL; Santa Cruz Biotechnology Inc.) and anti-CD68 (1:3200; 0.5 mg/mL; Santa Cruz Biotechnology Inc.) antibodies and rabbit polyclonal anti-FoxP3 (1:100; 0.2 mg/mL; Santa Cruz Biotechnology Inc.) and anti-IL10 (1:50; 0.2 mg/mL; Santa Cruz Biotechnology Inc.) antibodies. Antigen retrieval was performed with citrate buffer solution, pH 6.0 (for CD3, CD20, FoxP3 and CD68) or pH 9.0 (for CD4, CD8 and IL-10), at 125˚C for 1 min in a pressure cooker (Pascal), and the samples were incubated with the primary antibody overnight at 4˚C. According to the manufacturer's instructions, the reaction was visualized using a biotin-streptavidin peroxidase kit (Vector). 3,3'-Diaminobenzidine (Sigma Chemical, St. Louis, MO) was used as the chromogen. The sections were counterstained with Harris hematoxylin (H&E; Merck, Darmstadt, Germany). Positive controls for the reaction were performed on rat lung tissue and are shown in S1 Fig.

## ELISA for anti-Col II and anti-Col V antibodies

The plates (Immunolon II, Thermo Fisher Scientific) were sensitized with Col II and Col V (1 μg/well), diluted in bicarbonate buffer, pH 9.6, and incubated overnight at 4˚C. The plates were washed with 0.05% PBS in Tween 20 (Sigma–Aldrich) and blocked with 1% BSA in PBS (Sigma–Aldrich) for 1 hour at room temperature. Serum samples were diluted 1:50 in 1% BSA in PBS with 0.05% Tween 20 (Sigma–Aldrich), added to duplicate wells, and incubated for 1 hour at room temperature. After a wash cycle, an alkaline phosphatase-conjugated goat anti-rat IgG antibody (Sigma–Aldrich) was diluted 1:1000 in 1% BSA in PBS with 0.05% Tween 20 and added to each well for 1 hour at room temperature. The reaction was visualized by adding 50 μl/well of 1 mg/mL p-nitrophenyl phosphate (pNPP) diluted in 1 M diethanolamine and 0.5 mM MgCl2 buffer, pH 9.8. Optical density was determined with an ELISA reader (Labsystem Multiskan MS) at 405 nm. Control serum was obtained from the rats before arthritis induction. The cutoff values for Col II and Col V antibodies were determined by adding the mean of the serum negative controls three times the standard deviation.

## Immunofluorescence analysis of collagen fibers

For Col I immunostaining, exposure and recovery of antigenic sites were performed in citrate buffer, pH 6 (Diagnostic BioSystems, Pleasanton, CA, USA), for 10 min at 95˚C in a steam

cooker. For immunostaining of Col III and V, digestion with bovine pepsin (8 mg/500 μl 0.5 N acetic acid) (Sigma Chemical Co., St. Louis, MO, USA; 250 units/mg) was performed for 30 minutes at 37˚C. Nonspecific binding sites were blocked by incubating the sections in 5% BSA in PBS for 30 minutes. Then, the slides were incubated overnight at 4˚C with rabbit polyclonal anti-Col I (1:100; 1.15 mg/mL; Rockland), anti-Col III (1:100; 1.16 mg/mL; Rockland) and anti-Col V (1:600) [27] antibodies diluted in PBS. After this period, the sections were incubated for 1 hour with Alexa 488-conjugated goat anti-rabbit IgG antibodies (Invitrogen, Life Technologies) diluted 1:200 in 0.006% Evans blue. Finally, the slides were mounted on coverslips with buffered glycerin solution and analyzed under a fluorescence microscope (Olympus BX-51, Olympus Co., Tokyo, Japan). Positive controls for the reaction were performed on rat skin and are shown in S2 Fig.

Briefly, for colocalization assays, slides were incubated overnight at 4˚C with a mouse monoclonal anti-IL10 antibody (1:50; 0.2 mg/mL; Santa Cruz Biotechnology Inc.) and a rabbit polyclonal anti-FoxP3 antibody (1:100; 0.2 mg/mL; Santa Cruz Biotechnology Inc.). After this period, the sections were incubated for 1 hour at room temperature with Alexa Fluor 488-conjugated (green) and Alexa 546-conjugated (red) anti-rabbit/mouse IgG antibodies (Invitrogen, Life Technology) diluted 1:200 in PBS. The cell nuclei were labeled by incubation with DAPI (Abcam, Cambridge, UK) for 5 minutes. The slides were analyzed under a fluorescence microscope (Olympus BX-51, Olympus Co., Tokyo, Japan).

## Cellular and collagen fiber quantification

Morphometric analysis was performed using Image-Pro Plus 6.0 software. The stereological point counting method [28] was used to quantify the immunostained cells and cytokines; a reticular grid with 100 points distributed orthogonally on the acquired image was used. Ten fields of synovial tissue were evaluated at a magnification of 1000×, and cell expression was determined according to the number of positive cells coinciding with the crosshair grid in each field and is expressed as the percentage of positive cells relative to the total number of cells. To quantify the area of synovial tissue occupied by collagen fibers, images were acquired at a magnification of 400×, and 10 fields of view were evaluated. The collagen fibers were evaluated by selecting the fluorescent green hue corresponding to each type of collagen stained. The immunostained area was divided by the total area of the analyzed tissue, and the result was expressed as a percentage.

## Cell culture and IL-10 production analysis

The spleens of the IA and IA-Col V groups were collected aseptically and maintained in RPMI 1640 medium supplemented with 10% fetal bovine serum and 1% penicillin for maceration through a metal mesh for cell separation. The single-cell suspension was maintained for 2 minutes to separate the particulate material. The supernatant was washed in RPMI 1640 by centrifugation at 800 ×g for 5 minutes at 4˚C. The mononuclear cells were separated by density gradient with Ficoll-Paque in sterile PBS for 30 minutes at 500 ×g and 20˚C. The layer of mononuclear cells was washed in RPMI 1640. After the cells were stained with Trypan blue, the viable cells were counted in a Neubauer chamber. Then, the cells were placed in a 96-well culture plate (R&D Systems, Minneapolis, MN, USA) with $5 \times 10^5$ cells in 300 μL of RPMI 1640 per well and stimulated with Col V (100 μg/mL), concanavalin A (5 μg/mL) or without stimulus. The cells were maintained in a humified chamber with 5% CO2 at 37˚C for 48 hours. The conditioned culture media were collected and maintained at -70˚C until analysis. Next, we measured the concentration of IL-10 by a capture ELISA kit (R&D Systems, Minneapolis, MN, USA) using duplicate samples according to the manufacturer's instructions.

## Statistical analysis

For statistical comparisons of the means/medians of the groups, one-way ANOVA and two-way ANOVA or the Kruskal–Wallis test were used, in addition to the Dunn or Sidak posttest. For correlations between variables, Spearman's test was used. When analyzing the differences between means/medians, a *p* value of less than 5% ($p < 0.05$) was considered to indicate statistical significance.

## Results

### Oral Col V therapy promotes vascular and fibrotic repair of the synovia

First, we evaluated whether oral Col V therapy results in different histologic phenotypes. The histologic examination of the CT-Col V control group right and left knees revealed a uniform synovial membrane, with a thin layer of synoviocytes resting on an extracellular matrix composed of connective tissue, adipose tissue, muscle and vessels (Fig 2A, top panel, S3 Fig). In contrast, the synovia from the right knee of the IA group exhibited damage to synoviocytes and inflammatory response rich in mononucleated cells diffusely distributed throughout the extracellular matrix (Fig 2A, middle panel). After oral tolerance to Col V, the inflammatory response in the synovia of IA-Col V animals was minimized along the extracellular matrix (Fig 2A, bottom).

### PET/CT scans and inflammation score

At this stage, we compared the histologic score to those of other appropriate scoring systems. We tested PET/CT as a tool for detecting the degree of inflammatory response in the synovia and confirmed that PET/CT is a powerful tool for detecting inflammation. However, no difference was observed between treated and untreated animals after 30 days (Fig 2C and 2D), due to the large dispersion in the radiotracer uptake in the animals. On the other hand, the histopathological Krenn score, used to evaluate the inflammatory response, was significantly lower in the IA-Col V group than in the IA group (2.2 ± 1.135 vs. 6.5 ± 1.841; $p < 0.0001$) (Fig 2B). Thus, no association was found between the PET score and Krenn´s score (Fig 3).

### Immune cell mapping in induced arthritis

The next step was to immunophenotype the mononucleated cells present in the inflammatory response. We found significantly fewer CD3+ T lymphocytes in the IA-Col V group than in the IA group (45.83 ± 9.218 vs. 62.34 ± 13.99; p = 0.0056); similarly, CD4+ T lymphocytes were reduced (23.02 ± 9.442 vs. 39.88 ± 16.72; p = 0.0050), as were CD20+ B lymphocytes (31.56 ± 13.41 vs. 53.60 ± 21.43; p = 0.0108) and CD68+ macrophages (5.791 ± 2.413 vs. 10.92 ± 4.684; p = 0.0137) in the IA-Col V group after oral immunotherapy to Col V compared to those in the nontreated IA group (Fig 4). There was no significant difference in CD8+ T lymphocytes between the IA-Col V and IA groups (Fig 4). A moderate correlation was observed between CD8+ T cells and Krenn´s score (Fig 3; ρ = 0.486, P = 0.024).

### Induced arthritis and autoimmunity to collagen types II and V

We tested the levels of anti-Col antibodies of types II and V. Both the IA and IA-Col V groups presented positivity for circulating anti-Col II and anti-Col V autoantibodies. As shown in Fig 5, there was no significant difference in the antibody level between the IA and IA-Col V groups.

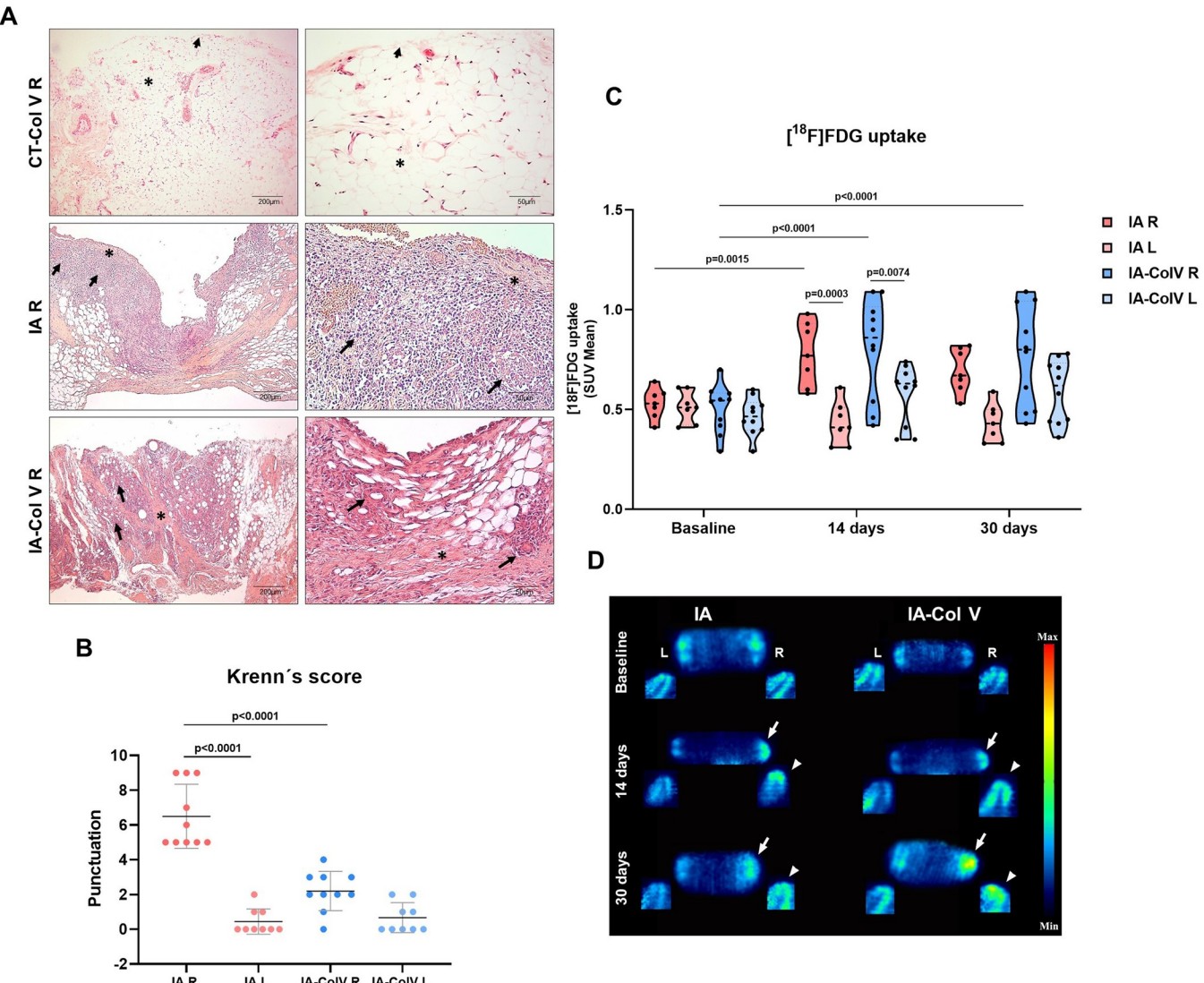

**Fig 2. Microscopic and [¹⁸F]FDG PET features of the joints in the IA and IA-Col V groups.** Panel A: Histological sections of synovial tissue stained with H&E from the IA, IA-Col V and CT-Col V groups (arrowhead: synovial membrane; arrow: immune cells; asterisk: collagen fiber deposition). Panel B: Graphical representation of Krenn's score based on histopathology analysis in the IA and IA-Col V groups. GraphPad Prism 8.0 software was used for one-way ANOVA with Sidak's posttest. Panel C: [¹⁸F]FDG uptake presented as the standardized uptake value (SUV). GraphPad Prism 8.0 software was used for two-way ANOVA with Sidak's posttest. Panel D: Illustrative [¹⁸F]FDG PET images of the right (R) and left (L) knees in the sagittal (arrow) and coronal planes (arrowhead) for the IA and IA-Col V groups.

## Col V oral therapy and increases in the numbers of IL-10+ and FoxP3 + cells

Then, we evaluated other immunomodulatory effects of Col V antigen treatment, such as anti-inflammatory IL-10 and immunosuppressive FoxP3. We found a significantly greater number of IL-10+ and FoxP3+ lymphocytes in the extracellular matrix in the IA-Col V group than in the IA group (59.35 ± 6.403 vs. 41.24 ± 17.20; p = 0.0009; 23.32 ± 6.490 vs. 11.85 ± 5.415; p = 0.0001, respectively) (Fig 6A and 6B). Furthermore, double immunostaining revealed increased colocalization of FoxP3+ and IL-10+ in lymphocyte cells in the IA-Col V group compared with the IA group (Fig 6C). In addition, we found a significant increase in IL-10 production after spleen cells stimulation with Col V *in vitro* between the IA-Col V and IA

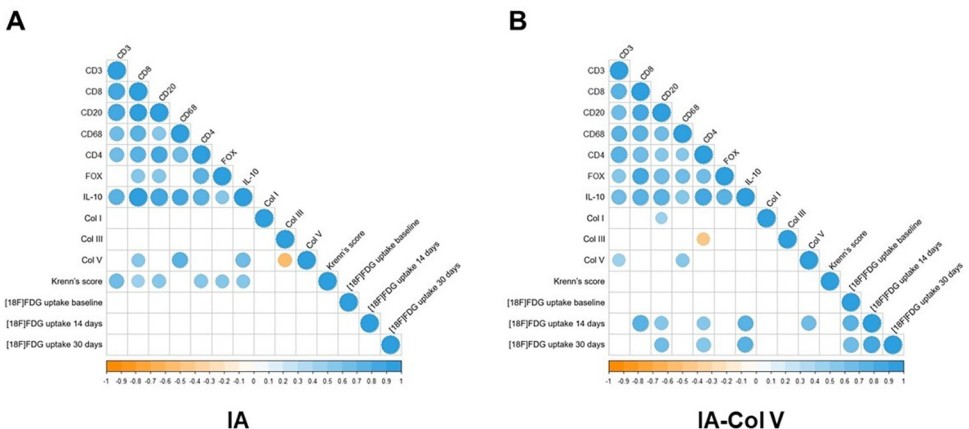

**Fig 3. Correlation between the variables in groups IA and IA-Col V.** The gradation of the colors represents a positive or negative correlation. The size of the dot represents Spearman's rho, and larger dots have values closer to 1, indicating a stronger correlation. IBM SPSS 22 software was used for Spearman´s test.

groups (p = 0.0070) (Fig 6D). Interestingly, a significant correlation was found between IL-10 + and FoxP3 (Fig 3B; ρ = 0.753, p<0.001) and between IL-10+ and [$^{18}$F]FDG uptake after 30 days in the IA-Col V group (Fig 3B; ρ = 0.739, P = 0.001).

## Oral administration of Col V and the repair/remodeling process in the injured synovia

Finally, we evaluated the effect of Col V therapy on fibrillar components of the extracellular matrix of the synovia. A significant increase in the green fluorescence of Col I was observed

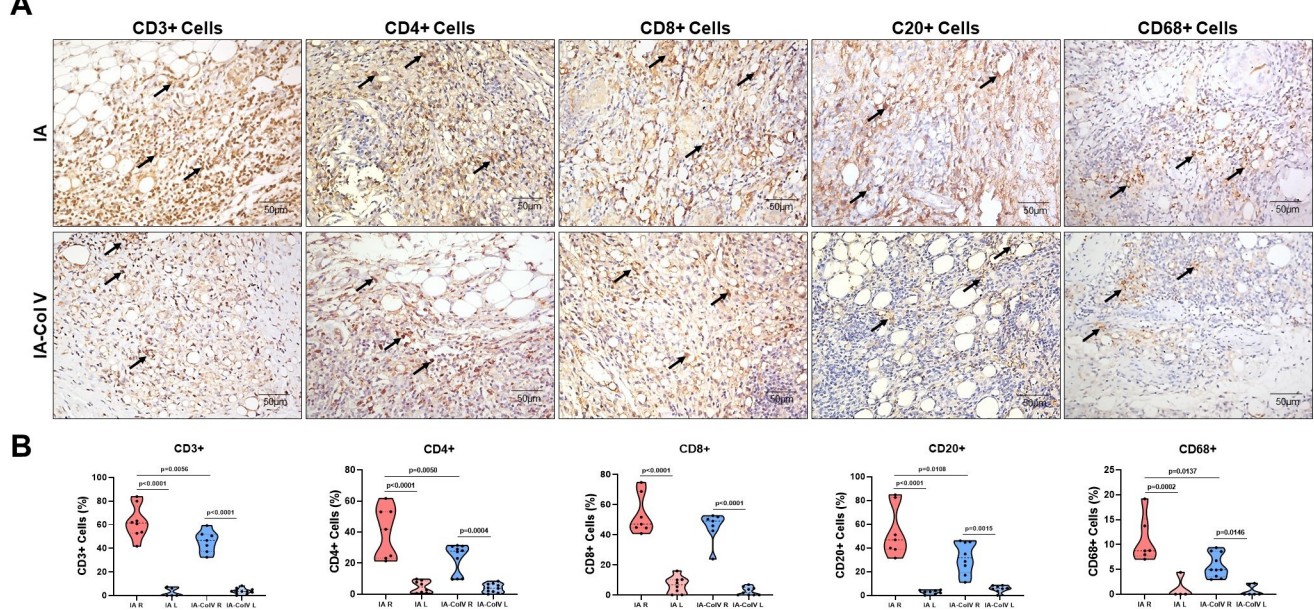

**Fig 4. Immunohistochemistry of synovial tissue from the right joint of the IA and IA-Col V groups.** Panel A: CD3, CD4, CD8, CD20 and CD68 expression in the subsynovial tissue (arrows) (original magnification: 400x). Panel B: Graphical representation of the amounts of CD3, CD4, CD8, CD20 and CD68 expressed in the synovial tissue. GraphPad Prism 8.0 software was used for one-way ANOVA with the Sidak posttest.

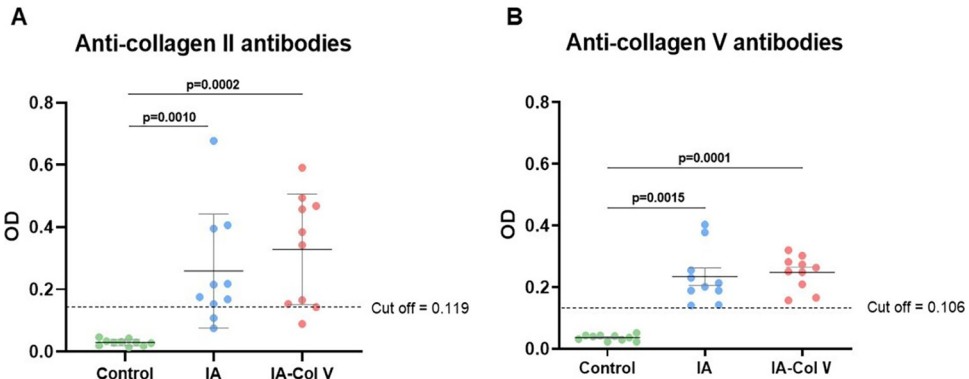

**Fig 5. Anti-collagen antibody frequency in the sera of the control, IA and IA-Col V groups.** Panels A and B show a graphical representation of types II (A) and V (B) anti-collagen antibody frequencies in the sera of animals from the control, IA and IA-Col V groups. We considered positivity to anti-Col II and V to optical density (OD) values above the cutoff (the cutoff line indicates OD values = 3 SD above the mean OD of 13 control serum samples obtained from Lewis rats before induced arthritis (on day zero)). SD: standard deviation. GraphPad Prism 8.0 software was used for the Kruskal–Wallis test with Dunn post hoc test.

around the vessel and synovial basement membrane in IA-Col V animals compared to IA animals (Fig 6E), which coincided with the quantitative evaluation of Col I in the IA-Col V group compared with the IA group (17.13 ± 3.395 vs. 11.15 ± 3.22; p = 0.0048; Fig 6F). Similarly, increased green fluorescence of Col V fibrils was found in close contact with immune cells in the IA group compared to the IA-Col V group (12.16 ± 5.85 vs. 2.88 ± 1.20; p<0.0001; Fig 6F). No significant difference in Col III expression was observed between the IA and IA-Col V synovia (Fig 6E and 6F).

## Discussion

In the present study, we investigated the potential effect of Col V oral immunotherapy on the synovial compartment of an mBSA/CFA-induced monoarthritis model. We demonstrated that oral administration of Col V post-intra-rticular injury attenuated the inflammatory response in the synovia and accelerated the repair/remodeling process of the injured synovia, which was mediated by an increase in FoxP3+ and IL-10+ in the synovial compartment and a decrease in the inflammatory process. Furthermore, the repair/remodeling in animals subjected to oral administration of Col V was characterized by increased deposition of Col I and decreased deposition of Col V.

The histological evaluation of arthritis on the 30th day suggested that the increased cell metabolism shown by the [18F]FDG PET images is a result of the intense infiltration of immune cells distributed throughout the synovial stroma, consisting mainly of CD4+, CD8+, CD20+ and CD68+ cells, in the arthritis model [29]. After oral immunotherapy with Col V, the numbers of CD3+, CD4+, CD20+ and CD68+ cells decreased in the IA-Col V group. Furthermore, in the short period that Col V immunotherapy was administered, there was a decrease in the severity of synovitis, as measured by Krenn´s score, suggesting that Col V immunotherapy modulated the inflammatory process leading to synovial tissue repair. Notably, in the synovial stroma of the IA-Col V group, a greater percentage of IL10+ cells were found in the cellular infiltrate. This finding combined with a greater number of FoxP3+/IL10 + cells suggests an immunoregulatory microenvironment characterized by increased FoxP3 + T regulatory cells (Tregs) and an immunosuppressive signature due to the production of the anti-inflammatory cytokine IL-10 [30–35]. Although there was no decrease in [18F]FDG PET

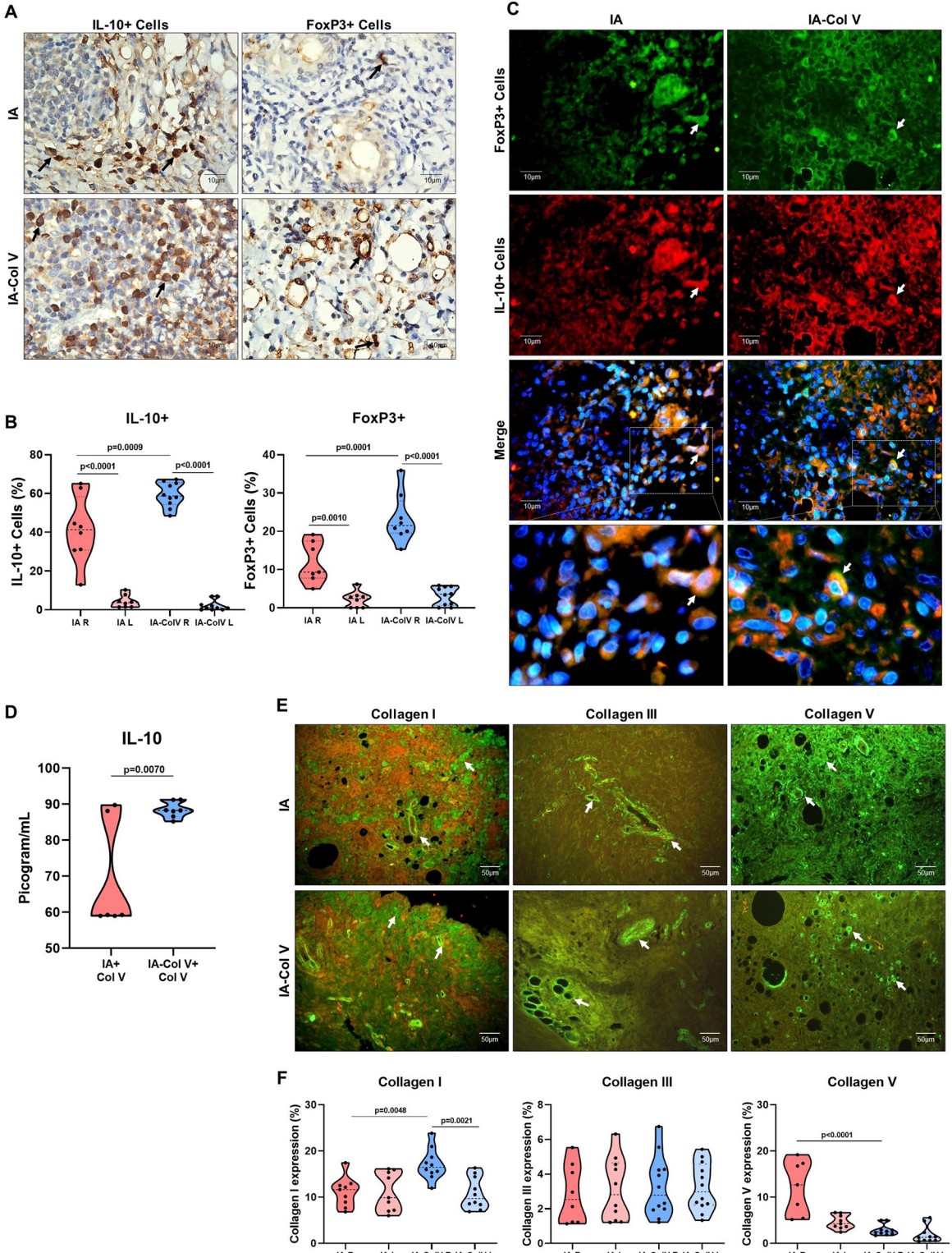

**Fig 6. Immunohistochemistry and immunofluorescence of synovial tissue from the IA and IA-Col V groups.** Panel A: IL-10 and FoxP3 expression in the subsynovial and perivascular regions (arrows). Panel B: Graphical representation of the amounts of IL-10 expressed in the synovial tissue. Panel C: Colocalization of immunofluorescence in the synovial tissue to FoxP3 (green), IL10 (red), FoxP3/IL10 (merge) and details of the merge. The nuclei were stained blue with DAPI. Panel D: Graphical representation of the IL-10 + dose in conditioned culture medium from IA-Col V and IA groups spleen cells stimulated by Col V. Panel E: Immunofluorescence of

collagen I, III and V (arrows). Panel F: Graphical representation of the amounts of Col I, III and V in the synovial tissue. GraphPad Prism 8.0 software was used for unpaired t tests.

images after oral immunotherapy with Col V, there was a strong correlation between IL-10 and FoxP3 and between IL-10 and [$^{18}$F]FDG uptake at 30 days.

In summary, these data revealed an increased presence of IL-10+ and FoxP3+ T cells as well as a decreased Krenn score in the group subjected to Col V oral immunotherapy. This finding was reinforced by *in vitro* assays employing lymphocytes isolated from the spleens of the IA and IA-Col V groups stimulated with Col V, which suggests T-cell activation and (de)polarization into immunosuppressive IL10+ T cells in the IA-Col V group. Notably, this finding suggested that oral immunotherapy with Col V results in increased IL-10 synthesis, which could act in the synovial repair and remodeling.

Interestingly, the synovial tissue of the IA-Col V group, which was characterized by high expression of IL-10-positive cells, had lower expression of Col V and higher expression of Col I, indicating that the anti-inflammatory microenvironment promoted the acceleration of tissue remodeling. Based on the literature, we speculate that increased IL-10 expression in synovial tissue after oral treatment with Col V possibly favors a fibrotic histological phenotype, with a predominance of subtypes of fibroblast-like synoviocytes with greater expression of Col I and lower expression of Col V. The differential expression of genes for the main synovial tissue proteins in subtypes of fibroblast-like synoviocytes has already been determined, and the positive subtype for CXCL14 stands out due to the greater expression of Col I in relation to that of Col V, emerging as an important candidate that would corroborate the expression of these proteins in the synovial tissue of the arthritis model treated orally with Col V [36]. However, *in vitro* studies involving the cultivation of fibroblast-like synoviocytes stimulated with IL-10 and the evaluation of the predominant synoviocyte subtypes, in addition to the expression of different types of collagen, are needed to test this hypothesis. On the other hand, the synovial stroma in the IA group had high expression of Col V surrounding the cell infiltrate, associated with reduced infiltration of FoxP3/IL-10+ cells, indicating a proinflammatory environment, which results in significantly more tissue damage and exacerbated tissue remodeling [34]. This suggests that immune cells can produce cytokines and growth factors, which stimulate the synthesis of Col V collagen by synovial fibroblasts in an attempt to repair tissue damage [37, 38].

In the present study, the arthritis induced by intraarticular injection of mBSA favored synovial damage and an inflammatory response in the presence of recruited immune cells. We speculate that during mBSA/CFA-induced monoarthritis, Col V becomes an autoantigen exposing Col V epitopes, which activates the cellular and humoral immune response leading to circulating anti-Col V autoantibodies [25]. In fact, we detected anti-Col V antibodies in the sera from both the IA and IA-Col V groups, suggesting a potential antigenic role of Col V in synovial inflammation. However, the Col V oral immunotherapy, which was maintained until the thirtieth day, did not change the serum anti-Col V profile during this period in the Col V-treated group. In our previous studies, circulating anti-Col V autoantibodies were also detected in a arthritis model, which supports our hypothesis that synovial inflammation exposes Col V epitopes, which activate the humoral immune response [25]. In fact, other studies have suggested that Col V has immunogenic and antigenic properties and can become an autoantigen when exposed to metalloproteinases or other agents that induce chronic tissue injury [8, 17, 18]. We also detected positivity for anti-Col II antibodies, which corroborates reports in the literature, since it has previously been detected in patients with arthritis and is related to joint destruction and the production of proinflammatory cytokines in the early stage of the disease [39–42].

According to our hypothesis that Col V becomes an autoantigen in mBSA/CFA-induced arthritis, we suggest that oral therapy with this protein could stimulate the regulatory mechanisms of the gastrointestinal mucosa. Consequently, it can trigger an immunoregulatory environment in synovial tissue and indirectly promote an anti-inflammatory effect in the [43]. We propose that Col V, when administered orally, passes through the digestive tract of animals, where some of the protein (peptides) is absorbed in the intestinal mucosa by dendritic cells, which migrate to the mesenteric lymph nodes and induce the differentiation of native T cells specific for Col V into Treg cells. In the intestinal lamina propria, under the influence of the large amount of IL-10, there is clonal expansion of Treg cells, which migrate to the site of injury, the synovial stroma, and release anti-inflammatory cytokines [33–35, 44, 45].

In this study, we characterized an immunosuppressive microenvironment with a high abundance of IL-10+ cells in synovial tissue after Col V oral therapy. However, it will also be important to evaluate the effect of Col V immunotherapy on the proinflammatory cytokines present in the mBSA/CFA-induced arthritis model, for example, by comparing Col V oral therapy with a known inflammatory/immunosuppressant. Another relevant point to be addressed is the time of initiation of immunotherapy. In this phase of the study, we chose to start treatment immediately after the last intra-articular injection of mBSA. However, new studies are needed to evaluate the effect of Col V oral immunotherapy at earlier time points.

We concluded that oral treatment with Col V in mBSA/CFA-induced arthritis attenuated damage to the synovium and accelerated synovial tissue repair and may be a potential immunotherapy for arthritis. In addition, *in situ* results showed that this process could be mediated by an increase in regulatory T FoxP3+/IL-10+ cells, although other studies are needed to confirm this hypothesis.

## Supporting information

**S1 Fig. Immunohistochemistry positive controls.** Immunostaining for CD3, CD4, CD8, CD20, FoxP3, and IL-10 (arrows) was performed on normal rat pulmonary tissue. Original magnifications: 400x left panel and 1000x right panel.
(TIF)

**S2 Fig. Positive controls for immunofluorescence staining.** Immunostaining for Col I, III and V was performed on normal rat skin. Original magnification: 400x.
(TIF)

**S3 Fig. Histological sections of synovial tissue.** H&E-stained synovium from the left (L) joint of the CT-Col V, IA, and IA-Col V groups. Original magnification: 40x left panel and 400x right panel.
(TIF)

**S1 Dataset. Analysis mean.** Mean of each analysis performed in IA and IA-Col V groups.
(PDF)

## Acknowledgments

We thank Suely Kunimi Kubo Ariga, MSc of Discipline of Clinical Emergency (LIM 51), Faculdade de Medicina da Universidade de Sao Paulo, who provided us with the antibodies to perform the ELISA.

## Author Contributions

**Conceptualization:** Ana Paula P. Velosa, Walcy R. Teodoro.

**Formal analysis:** Lizandre Keren Ramos da Silveira, Caroline Cristiano Real, Amanda Flores Yanke, Zelita Aparecida de J. Queiroz, Vitória Elias Contini, Thays de Matos Lobo, Camila Machado Baldavira, Walcy R. Teodoro.

**Funding acquisition:** Walcy R. Teodoro.

**Investigation:** Lizandre Keren Ramos da Silveira, Ana Paula P. Velosa, Walcy R. Teodoro.

**Methodology:** Lizandre Keren Ramos da Silveira, Ana Paula P. Velosa, Sergio Catanozi, Marco Aurélio A. Pereira, Antonio dos Santos Filho, Fabio Luiz N. Marques, Daniele de Paula Faria, Caroline Cristiano Real, Sandra de M. Fernezlian, Amanda Flores Yanke, Thays de Matos Lobo, Solange Carrasco, Cláudia Goldenstein-Schainberg, Walcy R. Teodoro.

**Project administration:** Walcy R. Teodoro.

**Resources:** Ana Paula P. Velosa, Sergio Catanozi, Antonio dos Santos Filho, Fabio Luiz N. Marques, Daniele de Paula Faria, Sandra de M. Fernezlian, Solange Carrasco, Cláudia Goldenstein-Schainberg, Vera L. Capelozzi, Walcy R. Teodoro.

**Supervision:** Walcy R. Teodoro.

**Validation:** Sergio Catanozi, Marco Aurélio A. Pereira, Antonio dos Santos Filho.

**Visualization:** Lizandre Keren Ramos da Silveira, Ana Paula P. Velosa, Zelita Aparecida de J. Queiroz, Vitória Elias Contini, Camila Machado Baldavira, Vera L. Capelozzi.

**Writing – original draft:** Lizandre Keren Ramos da Silveira, Ana Paula P. Velosa, Walcy R. Teodoro.

**Writing – review & editing:** Ana Paula P. Velosa, Sergio Catanozi, Fabio Luiz N. Marques, Daniele de Paula Faria, Ricardo Fuller, Vera L. Capelozzi, Walcy R. Teodoro.

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
