## [Decision Letter · Decision Letter 0]

16 Jul 2024

PONE-D-24-19996Immunotherapeutic potential of collagen V oral administration on synovitis in mBSA/CFA-induced arthritisPLOS ONE

Dear Dr. Teodoro,

Thank you for submitting your manuscript to PLOS ONE. After careful consideration, we feel that it has merit but does not fully meet PLOS ONE’s publication criteria as it currently stands. Therefore, we invite you to submit a revised version of the manuscript that addresses the points raised during the review process.

Based on the reviewers' suggestions, the paper needs major revision.  The reviewers' comments can be found below.

We look forward to receiving your revised manuscript.

Kind regards,

Tanja Grubić Kezele, Ph.D., M.D.

Academic Editor

PLOS ONE

Journal Requirements:

3. Thank you for stating the following financial disclosure: "This work was supported by the São Paulo Research Foundation - FAPESP [2019/24178-0]."  

4. Please expand the acronym “FAPESP” (as indicated in your financial disclosure) so that it states the name of your funders in full.

Reviewers' comments:

Reviewer's Responses to Questions

**Comments to the Author**

1. Is the manuscript technically sound, and do the data support the conclusions?

Reviewer #1: Partly

2. Has the statistical analysis been performed appropriately and rigorously? 

Reviewer #1: Yes

3. Have the authors made all data underlying the findings in their manuscript fully available?

Reviewer #1: No

4. Is the manuscript presented in an intelligible fashion and written in standard English?

Reviewer #1: Yes

5. Review Comments to the Author

Reviewer #1: This study investigates the effect of collagen V oral administration on the synovitis. The authors induced mBSA/CFA-induced arthritis in rats and compared several factors among three groups, some with oral administration of collagen V and others without. While this study may be interesting, I have some major comments.

1. Why can mBSA/CFA-induced arthritis be used to study the synovitis? Synovitis and arthritis are not exactly the same. Add more contents on their relationship between them.

2. Explain why IA is the abbreviation of arthritis.

3. The quality of figures needs significantly improved.

4. You consider three groups , IA, IA-COl V, and CT-Col. But you do not consider only CT group without oral administration of collagen.

5. The result shows that IA-COl V group have reduction in Col V and an

increase in Col I in the synovia compared to those in the IA group. Why is there oral administration in the IA-Col V group, yet Col V is reduced in this group? And why is there an increase in Col I?

6. Add some background of the relationship between Col V and Col I.

6. PLOS authors have the option to publish the peer review history of their article (what does this mean?). If published, this will include your full peer review and any attached files.

Reviewer #1: No

---

## [Author Response · Author response to Decision Letter 0]

2 Sep 2024

Dear reviewers, 

Firstly, we would like to thank the Reviewer and editor for the critical comments that allowed us to rewrite the work to make it more fluent and clearer. After several readings of the criticisms made by the Reviewer, we noticed that the work could be scientifically improved thus, we re-evaluated the Title, Introduction and Figures. In this way, we would appreciate it if the reviewer could re-read the new manuscript for a new opportunity to be accepted by PLOS ONE. We revised the manuscript based on these comments, we are re-submitting the paper with tracked changes and revised paper without tracked changes, along with a point-by-point response to the reviewer's comments. In addition, in the new version of the manuscript, we write the of “Data availability statement”. The intellectual exercise carried out between authors and reviewers undoubtedly increased the scientific value of the work. Thank you very much.

Editor's Considerations

Response: Thank you for the suggestion. We deposit our laboratory protocols in protocols.io (DOI: dx.doi.org/10.17504/protocols.io.81wgbz9d1gpk/v1).

Journal Requirements:

Query 1. Please ensure that your manuscript meets PLOS ONE's style requirements, including those for file naming. The PLOS ONE style templates can be found at 

Response: Yes, our manuscript meets PLOS ONE's style requirements.

Query 2. To comply with PLOS ONE submissions requirements, in your Methods section, please provide additional information regarding the experiments involving animals and ensure you have included details on (1) methods of sacrifice, (2) methods of anesthesia and/or analgesia, and (3) efforts to alleviate suffering.

Response: In the new version of the manuscript, we provided additional information regarding (1) methods of sacrifice, (2) methods of anesthesia and/or analgesia, and (3) efforts to alleviate suffering. Thank you for the observation.

“For preventive analgesia, tramadol hydrochloride injections (40 mg/kg body weight) were administered subcutaneously one hour before anesthesia and every 8 hours during 48 hours after the experimental procedure. Intraperitoneal anesthesia (ketamine hydrochloride at a dose of 100 mg/kg body mass and xylazine hydrochloride at a dose of 10 mg/kg body mass) was administered”

“The animals were euthanized by intraperitoneal injection of an anesthetic overdose (ketamine hydrochloride at a dose of 300 mg/kg body mass and xylazine hydrochloride at a dose of 30 mg/kg body mass).”

Query 3. Thank you for stating the following financial disclosure: "This work was supported by the São Paulo Research Foundation - FAPESP [2019/24178-0]." 

Response 3: Thank you for your comment. The financial disclosure was changed in the new version of the manuscript:

“This work was supported by grants from the São Paulo Research Foundation - FAPESP [2019/24178-0] for the Silveira LKR, FAPESP [2018/20403-6 and 2023/02755-0] for Capelozzi, VL and Baldavira CM, FAPESP [2021/13220-5] for Velosa AP and from CNPq - National Council for Scientific and Technological Development [303735/2021-0] for Capelozzi, VL. The FAPESP is a public institution that promotes academic research of the government of the state of São Paulo, Brazil. The CNPq is an organization linked to the Ministry of Science, Technology and Innovation to encourage research in Brazil. The funders had no role in study design, data collection and analysis, decision to publish, or preparation of the manuscript.”

Query 4. Please expand the acronym “FAPESP” (as indicated in your financial disclosure) so that it states the name of your funders in full.

Response: The São Paulo State Research Support Foundation (FAPESP) is a public institution that promotes academic research of the government of the state of São Paulo, Brazil. The CNPq is an organization linked to the Ministry of Science, Technology and Innovation to encourage research in Brazil.

Reviewer 1

Query 1. Why can mBSA/CFA-induced arthritis be used to study the synovitis? Synovitis and arthritis are not exactly the same. Add more contents on their relationship between them.

Response: Yes. The reviewer's considerations are very pertinent, since we did not clearly explain why we used this arthritis model. In fact, this is an arthritis model and in the title we tried to include the joint component that was the target of our study. In the new version of the manuscript, we changed the title and the term synovitis was excluded. We still know that in autoimmune or infectious arthritis and in other forms of arthritis, the most affected tissue is the synovial tissue, however other joint components are affected. The synovitis is the main process triggered by the disease, but they develop in different ways. The choice to continue our study with this model is also due to our previous publication, (see Atayde, 2018), cited in our article, where we standardized the induction of joint inflammation with synovial involvement, in the animal's right knee, so that we could evaluate whether tolerance with Collagen V would act positively on the synovial inflammatory process. Furthermore, to clarify the reviewer, we know that the synovitis process is present in several osteoarticular diseases and that these also present synovitis, with similar characteristics, i.e. thickening of the synovial membrane and the formation of “pannus”, rich in infiltration of immune system cells and edema.

Query 2. Explain why IA is the abbreviation of arthritis.

Response: The IA is the abbreviation for Induced Arthritis. Thank you for your comment.

Query 3. The quality of figures needs significantly improved.

Response: The reviewer's considerations are right. In the new version of the article we improved significantly the quality of the figures. Thank you for the observation.

Query 4. You consider three groups, IA, IA-COl V, and CT-Col. But you do not consider only CT group without oral administration of collagen.

Response: We appreciate this comment very much. In our study, we did not include a CT group, since that the left contralateral side of the animals as the control didn´t show histological changes. In addition, we also respected the ethical committee for use of experimental animals that propose a reduction of the number of animals in the experimental protocol.

Query 5. The result shows that IA-COl V group have reduction in Col V and an

increase in Col I in the synovia compared to those in the IA group. Why is there oral administration in the IA-Col V group, yet Col V is reduced in this group? And why is there an increase in Col I?

Response: We appreciate this comment very much. In the Discussion wrting our hypothesis:

“Interestingly, the synovial tissue of the IA-Col V group, which was characterized by high expression of IL-10-positive cells, had lower expression of Col V and higher expression of Col I, indicating that the anti-inflammatory microenvironment promoted the acceleration of tissue remodeling. Based on the literature, we speculate that increased IL-10 expression in synovial tissue after oral treatment with Col V possibly favors a fibrotic histological phenotype, with a predominance of subtypes of fibroblast-like synoviocytes with greater expression of Col I and lower expression of Col V. The differential expression of genes for the main synovial tissue proteins in subtypes of fibroblast-like synoviocytes has already been determined, and the positive subtype for CXCL14 stands out due to the greater expression of Col I in relation to that of Col V, emerging as an important candidate that would corroborate the expression of these proteins in the synovial tissue of the monarthritis model treated orally with Col V (Kang et al., 2022). However, in vitro studies involving the cultivation of fibroblast-like synoviocytes stimulated with IL10 and the evaluation of the predominant synoviocyte subtypes, in addition to the expression of different types of collagen, are needed to test this hypothesis. On the other hand, the synovial stroma in the IA group had high expression of Col V surrounding the cell infiltrate, associated with reduced infiltration of FoxP3/IL-10+ cells, indicating a proinflammatory environment, which results in significantly more tissue damage and exacerbated tissue remodeling (Tordesillas; Berin, 2018). This suggests that immune cells can produce cytokines and growth factors, which stimulate the synthesis of Col V collagen by synovial fibroblasts in an attempt to repair tissue damage (Liu; Zhang, 2008; Zhou et al., 2021).”

Query 6. Add some background of the relationship between Col V and Col I.

Response: Many thanks for your suggestion. In the Introduction of the manuscript, we included a background about the relationship between collagen types. 

“Collagen V (Col V) is a glycoprotein accounting for 2 to 5% of all fibrillar collagen types. Collagen I (Col I), III (Col III) and Col V aggregate to form fibrillar structures, which are stabilized by intermolecular cross-links, resulting in the formation of a stable network, forming heterotypic fibrils (I/III/V) (Adachi; Hayashi; Hashimoto, 1991; Birk et al., 1988, 1990; Gelse; Pöschl; Aigner, 2003; Konomi et al., 1984; Linsenmayer; Fitch; Birk, 1990). The triple helix of Col V, formed by two alpha 1 chains (α1) and one alpha 2 chain (α2), is hidden between the Col I and Col III fibers. Unlike other types of collagen, Col V retains its amino terminal domain, which projects to the surface of the heterotypic fiber (Birk, 2001; Birk et al., 1990). As these domains reach a critical concentration, the aggregation of new collagen molecules to the surface becomes less favorable than the formation of new fibrils (Birk, 2001; Birk et al., 1990). The Col V is essential for the formation of the collagen network, as well as the control of fibrogenesis and the regulation of fiber size (Andrikopoulos et al., 1995; Asgari et al., 2017; Chanut-Delalande et al., 2004; Gelse; Pöschl; Aigner, 2003; Kadler; Hill; Canty-Laird, 2008; Mares et al., 2000). Furthermore, to its function of controlling the diameter of heterotypic fibers, Col V has other biological properties and in this sense, it has been demonstrated that a culture medium rich in Col V inhibits the growth and proliferation of endothelial cells (Fukuda et al., 1988; Horrocks; Ziats, 1993).”

---

## [Decision Letter · Decision Letter 1]

8 Sep 2024

Immunotherapeutic potential of collagen V oral administration in mBSA/CFA-induced arthritis

PONE-D-24-19996R1

Dear Dr. Teodoro,

We’re pleased to inform you that your manuscript has been judged scientifically suitable for publication and will be formally accepted for publication once it meets all outstanding technical requirements.

Kind regards,

Tanja Grubić Kezele, Ph.D., M.D.

Academic Editor

PLOS ONE

Additional Editor Comments (optional):

Reviewers' comments:

Reviewer's Responses to Questions

**Comments to the Author**

1. If the authors have adequately addressed your comments raised in a previous round of review and you feel that this manuscript is now acceptable for publication, you may indicate that here to bypass the “Comments to the Author” section, enter your conflict of interest statement in the “Confidential to Editor” section, and submit your "Accept" recommendation.

Reviewer #1: All comments have been addressed

2. Is the manuscript technically sound, and do the data support the conclusions?

Reviewer #1: Yes

3. Has the statistical analysis been performed appropriately and rigorously? 

Reviewer #1: Yes

4. Have the authors made all data underlying the findings in their manuscript fully available?

Reviewer #1: No

5. Is the manuscript presented in an intelligible fashion and written in standard English?

Reviewer #1: Yes

6. Review Comments to the Author

Reviewer #1: The authors have addressed most of my comments. I do not have further comments. Therefore, I suggest accepting this paper.

7. PLOS authors have the option to publish the peer review history of their article (what does this mean?). If published, this will include your full peer review and any attached files.

Reviewer #1: **Yes: **Hsiuying Wang

---

## [Editor Report · Acceptance letter]

27 Sep 2024

PONE-D-24-19996R1 

PLOS ONE

Dear Dr. Teodoro, 

I'm pleased to inform you that your manuscript has been deemed suitable for publication in PLOS ONE. Congratulations! Your manuscript is now being handed over to our production team.

Kind regards, 

on behalf of

Prof. dr. Tanja Grubić Kezele 

Academic Editor

PLOS ONE